# Using AI-Based Virtual Companions to Assist Adolescents with Autism in Recognizing and Addressing Cyberbullying

**DOI:** 10.3390/s24123875

**Published:** 2024-06-15

**Authors:** Robinson Ferrer, Kamran Ali, Charles Hughes

**Affiliations:** Synthetic Reality Lab, Department of Computer Science, University of Central Florida, Orlando, FL 32816, USA; robinson.vasquezferrer@ucf.edu (R.F.); kamran.ali@ucf.edu (K.A.)

**Keywords:** cyberbullying, natural language processing, language models, machine learning, Autism Spectrum Disorder (ASD)

## Abstract

Social media platforms and online gaming sites play a pervasive role in facilitating peer interaction and social development for adolescents, but they also pose potential threats to health and safety. It is crucial to tackle cyberbullying issues within these platforms to ensure the healthy social development of adolescents. Cyberbullying has been linked to adverse mental health outcomes among adolescents, including anxiety, depression, academic underperformance, and an increased risk of suicide. While cyberbullying is a concern for all adolescents, those with disabilities are particularly susceptible and face a higher risk of being targets of cyberbullying. Our research addresses these challenges by introducing a personalized online virtual companion guided by artificial intelligence (AI). The web-based virtual companion’s interactions aim to assist adolescents in detecting cyberbullying. More specifically, an adolescent with ASD watches a cyberbullying scenario in a virtual environment, and the AI virtual companion then asks the adolescent if he/she detected cyberbullying. To inform the virtual companion in real time to know if the adolescent has learned about detecting cyberbullying, we have implemented fast and lightweight cyberbullying detection models employing the T5-small and MobileBERT networks. Our experimental results show that we obtain comparable results to the state-of-the-art methods despite having a compact architecture.

## 1. Introduction

Bullying, a multifaceted societal dilemma of intricate dimensions, has undergone significant evolution in the contemporary era, particularly with the advent of technology and the pervasive influence of social media. A particularly pernicious manifestation of this challenge is recognized as cyberbullying, encompassing the malevolent deployment of digital technology to target and victimize individuals, thereby resulting in diverse forms of harm.

Statistics indicate that approximately 18% of children in Europe have been affected either by bullying or harassment through online and mobile communication channels. The EU Kids Online Report of 2014 revealed that close to 20% of children aged 11 to 16 are susceptible to cyberbullying [1]. A cyberbullying research team conducted a survey between July and October 2016 involving high school students. The results indicated that 34% of students had experienced cyberbullying at some point in their lives [2]. Similarly, according to the Pew Research Center [3], approximately two-thirds of adolescents in the USA have experienced cyberbullying. According to surveys from various sources, including Statista [4], Pew Research Center [5], and the Office for National Statistics [6], cyberbullying has been prevalent across different age groups and regions. In the USA, 41% of adults reported experiencing cyberbullying [4], while 46% of teens aged 13 to 17 have encountered cyberbullying [5]. In England and Wales, the Office for National Statistics found that 19% of children aged 10 to 15, equivalent to 764,000 children, have been victims of cyberbullying [6]. All of these instances underscore the critical need for identifying a suitable, efficient, and proven approach to addressing this online epidemic.

Previous research indicates that cyberbullying (CB) is associated with various adverse outcomes: anxiety [7,8,9], depression [10,11], social isolation [12], suicidal ideation [13,14], and self-harm [15,16]. According to Hinduja et al. [17], individuals targeted by cyberbullying often express discomfort or fear attending school, which negatively affects their academic performance. Additionally, nearly 70% of teenagers who experienced cyberbullying reported a decline in self-esteem, while nearly one-third noted an impact on their friendships. This highlights the necessity for establishing a resilient system to mitigate the dissemination of bullying content across online forums, blogs, and social media platforms, thereby addressing its societal impact.

Cyberbullying is often misunderstood, resulting in the creation of ineffective systems with limited practical utility. Moreover, many studies have solely focused on filtering cyberbullying using profanity, which represents only a single aspect of this issue. Profanity may not consistently signify bullying, especially on platforms predominantly used by young individuals [18,19]. Therefore, developers and media managers benefit from having a robust system that comprehends context more effectively to improve cyberbullying detection. Numerous machine learning (ML) algorithms have been suggested for this objective. However, their performance is inconsistent due to challenges in capturing the complete context, addressing the high class imbalance issue, and achieving generalization. In recent years, large language models (LLMs) such as BERT [20] and RoBERTa [21] have attained state-of-the-art (SOTA) results across various natural language processing (NLP) tasks. Unfortunately, the LLMs have not been applied extensively to detect cyberbullying in online web-based systems due to their large model size and slow inference time. In this paper, we explored the use of lightweight versions of these models for cyberbullying (CB) detection in a real-time online web-based system.

Within the broader landscape of those vulnerable to cyberbullying, it is imperative to direct focused attention toward adolescents with disabilities, specifically those situated on the autism spectrum. For this distinctive community, discerning instances of cyberbullying within the intricate realms of digital interactions often presents a formidable challenge. The primary objective of this paper resides in facilitating the acquisition of cyberbullying identification proficiencies among adolescents within the autism spectrum. This endeavor culminates in the creation of a virtual environment meticulously designed to present scenarios emblematic of cyberbullying situations. This web-based platform serves as a secure and immersive space where adolescents can acquaint themselves with the nuances of cyberbullying while fostering a comfortable learning atmosphere. The web-based system commences by exposing adolescents to a cyberbullying scenario and subsequently soliciting their insights to ascertain their aptitude in identifying instances of bullying. A virtual environment scenario having the virtual characters, the AI driven virtual companion, Maria (in the second row and second column), and the participant is shown in Figure 1.

The virtual setting is designed to replicate a cyberbullying scenario, where one virtual character bullies another. Following the cyberbullying scenario, the AI virtual companion prompts the participant to describe what they witnessed and whether they recognized cyberbullying. To assess the participant’s understanding of cyberbullying and their ability to identify such behavior automatically, we frame the task as text-based cyberbullying detection. While the participant’s response technically does not match instances of bullying speech, we posit that their description of the observed virtual environment can serve as input for the cyberbullying detection model. This framework also minimizes random guesses by the participant in cyberbullying detection. This innovative problem formulation, tailored for a unique and crucial application, constitutes a significant contribution of this paper. Our experimental findings indicate that this formulation enables our cyberbullying model to effectively ascertain whether the participant comprehends cyberbullying. Additionally, it indicates that the detailed responses provided by the participants largely contain the information necessary to achieve this objective.

While the corpus of information presents various modalities, including visual, auditory, and textual cues, this paper focuses exclusively on the analysis of textual information, with the intention of exploring other modalities in future investigations. Our research involves the development of an automatic cyberbullying detection model to assess the accuracy of adolescents’ identifications of bullying instances through the examination of textual information. However, due to constraints imposed by the data gathered from virtual sessions (which we call KIDS-cyberbullying dataset)—characterized by their limited scale and inherent imbalances—we have undertaken a preliminary phase involving pre-training on a substantially more extensive dataset. The dataset utilized for this pre-training stage originates from the Kaggle cyberbullying dataset [22], aggregating data from eight distinct cyberbullying repositories across a spectrum of social media platforms, each illustrating distinct forms of online harassment.

Moreover, considering our aspiration to deploy the model within a web-based system, we prioritize the implementation of lightweight models. Although larger models may offer improved performance, the need for compatibility with the web-based system necessitates the utilization of models that exhibit efficiency and nimbleness. Our research process involves fine-tuning the models on an augmented rendition of the original dataset, incorporating various data augmentation methodologies. To assess the efficacy of our approach, we conduct a rigorous comparative analysis against state-of-the-art models.

We develop our cyberbullying model by incorporating lightweight encoders such as T5-small [23] and MobileBERT [24] and compare our methods with the state-of-the-art large models such as BERT [20], RoBERTa [21], T5-base [23], and SimCSE [25]. We also apply data augmentation techniques such as back-translation [26] and paraphrasing [27] to increase the number of training samples. Notably, our results are found to be comparable to those of state-of-the-art models. Furthermore, we demonstrate that deploying lightweight models like MobileBERT and T5-small [23] incurs only a minimal performance decline, thus aligning with the requirements of our web-based system. This paper proceeds by providing detailed descriptions of our lightweight models in our web-based system deployed for cyberbullying detection. Subsequently, it delineates our methodologies for processing textual data from the Kaggle dataset, elucidates our data augmentation techniques applied to the original dataset, and expounds upon our fine-tuning procedures. Following this, we present our results and offer a comparative analysis vis-à-vis other state-of-the-art models. Finally, we conclude with discussions on avenues for future research and the incorporation of this research into a multi-modal bullying detection system capable of automating video annotations.

## 2. Related Work

Nowadays, adolescents have been more involved in online activities, especially on social networking platforms, which has elevated their vulnerability to cyberbullying. Therefore, researchers have devised automated techniques for detecting cybercrime, which encompasses the posting of irrelevant comments, offensive language, and threatening messages. For instance, Agarwal et al.’s [28] initial research delved into Twitter sentiment analysis, devising a model for two classification tasks: (1) Binary categorization into positive and negative sentiments, and (2) Three-way classification encompassing good, negative, or neutral sentiments. They formulated a tree representation of tweets to effectively consolidate various feature categories. The fusion of unigram with the senti-feature model surpassed their previous tree kernel-based model with a unigram baseline by over 4%. Nandhini et al. [29] introduced a system leveraging machine learning advancements, which centered on employing the Levenshtein algorithm to identify cyberbullying terms within a conversation. The naïve Bayes classifier was then utilized to categorize instances of cyberbullying. Singh et al. [30] introduce a probabilistic information integration framework that incorporates confidence ratings and interconnections associated with diverse social and textual elements. Each feature undergoes independent analysis through an early fusion approach, while the confidence levels of these features remain undifferentiated. Similarly, Raj et al. [31] investigated eleven classification algorithms and seven feature extraction methods across two datasets. The study analyzed the performance of algorithmic tasks utilizing feature extraction and word embedding techniques. A significant finding indicates that attention models, when combined with bidirectional neural networks, achieve high classification accuracy. Logistic regression emerged as the most efficient among traditional machine learning classifiers. In contrast, TF-IDF consistently delivered high accuracy levels. Deep neural networks outperformed existing algorithms in cyberbullying detection, achieving accuracy and F1 scores of 95% and 98%, respectively.

As interest and research in neural networks continue to grow, various deep learning techniques are being utilized for cyberbullying detection. Al-Ajlan et al. [32] introduced a method aimed at bypassing the phases of feature extraction and selection to enhance Twitter cyberbullying detection. The approach involves preserving tweet semantics through the substitution of tweets with word vectors. Subsequently, an optimization algorithm was employed to fine tune the parameters of the convolutional neural network (CNN), with the authors striving to achieve values as close to optimal as feasible. Banerjee et al. [33] utilized a convolutional neural network (CNN) approach with multiple layers, attaining an accuracy of 93.97%. Jain et al. [34] employed natural language processing (NLP) and machine learning algorithms to identify and categorize two prevalent forms of cyberbullying: personal attacks and hate speech observed on Twitter and Wikipedia. By employing three feature extraction methods (Word2Vec, TF-IDF, and Bag of Words) and four classifiers, the model attained an accuracy of 90% for Twitter data and 80% for Wikipedia data (using random forest, logistic regression, support vector machine, and multilayer perceptron). Saravanaraj et al. [34] proposed the Embeddings-augmented Bag-of-Words model, known as EBoW, as a method for cyberbullying detection. This model assigns different weights to a set of predefined insulting words based on word embeddings to extract bullying indicators. By integrating Bag-of-Words traits, latent semantic features, and bullying characteristics, EBoW achieved a precision rate of 76.8%. Unfortunately, the application of large language models (LLMs) for detecting cyberbullying in online web-based systems has been limited by their substantial model size and slow inference time. This paper investigates the utilization of lightweight variants of these models for cyberbullying (CB) detection in an online web-based system.

## 3. Methodology

### 3.1. Data Collection in a Virtual Environment

The virtual environment is set up to simulate a cyberbullying scenario, where one virtual character (CJ, depicted in a white shirt in the first row and second column of Figure 1) bullies another virtual character (Martin, shown in a green shirt in the bottom left of Figure 1). Martin’s friends, Ed and the AI virtual companion Maria (appearing in a black shirt in the second row and second column of Figure 1), assist Martin and try to prevent him from sharing his personal information online. Once the cyberbullying scenario concludes, Maria prompts the participant to describe their observations and whether they identified any instances of cyberbullying.

To automatically assess whether the participant understands cyberbullying and can recognize such behavior, we frame the task as text-based cyberbullying detection. Although the participant’s response technically does not correspond to instances of bullying speech, we hypothesize that the description of the virtual scenario observed by the participant can serve as input for the cyberbullying detection model. Employing such a framework also helps prevent random guesses by the participant regarding cyberbullying detection.

This innovative problem formulation, customized for a novel and critical application, stands out as one of the key contributions of this paper. Our experimental results demonstrate that, with this formulation, our cyberbullying model effectively fulfills its objective of determining whether the participant possesses knowledge of cyberbullying. Moreover, it suggests that the detailed responses provided by the participants predominantly contain the information required to accomplish this goal.

To develop the cyberbullying detection model, we first pre-train the network on a publicly available text-based cyberbullying detection dataset. Subsequently, we fine tune the model on the KIDS-cyberbullying dataset, which we compiled during practice sessions, wherein the virtual character Maria (transformed into an AI virtual companion) is controlled by a human interactor. Following pre-training and fine tuning, the cyberbullying detection model is deployed in the web-based system to detect if the participant has recognized cyberbullying during the virtual environment scenario, based on their response. The flowchart of the sequence of events in a virtual cyberbullying setup is shown in Figure 2. Specifically, text information is extracted from the participant’s video/audio response using automatic speech recognition. The extracted text is then inputted into the cyberbullying detection network to determine if the participant has identified cyberbullying. If the model indicates that the participant has detected cyberbullying, they are presented with an instructional video demonstrating how to safeguard their personal information in an online interactive environment, along with instructions on managing a cyberbullying scenario. Conversely, if the model suggests that the participant has not recognized cyberbullying, they are presented with video instructions detailing what cyberbullying entails and how to identify it. Following the presentation of the educational video on cyberbullying, the participant is reintroduced to the virtual environment setup, and the entire process is reiterated until the participant is educated not only on recognizing cyberbullying but also on how to address such situations effectively in an online context.

### 3.2. Data Pre-Processing

Responses were collected and processed by our web-based system, which extracts textual data from the videos of the participants using an automatic speech recognition technique named Whisper [35], an OpenAI audio-to-text model for cyberbullying detection. A notable limitation encountered was the scarcity of data, prompting the adoption of pre-training as a strategy to mitigate the constraints imposed by limited datasets.

### 3.3. Pre-Training on External Dataset

Given that we had a very limited KIDS-cyberbullying dataset from our web-based platform, we performed pre-training on a large dataset. We selected the Kaggle cyberbullying dataset [22], which encompasses a diverse array of cyberbullying instances from various social media platforms, categorically labeled as either containing or not containing bullying. This amalgamation of data from all these social media sources yielded a comprehensive dataset of 448,874 samples, constituting a sizable foundation for our pre-training phase. Our methodology incorporated the use of transformer-based encoders together with a linear classification layer to be trained on this dataset. However, an imbalance in the dataset was observed, with approximately 87% labeled as non-bullying and about 23% as bullying. To address this imbalance, data augmentation techniques were employed, specifically paraphrasing and back-translation.

### 3.4. Data Augmentation Techniques

#### 3.4.1. Back-Translation

Back-translation involves translating some text x from an original language L to a target Language L’ using model f, and then using another model g to translate from L’ back to L [36]. The hope is that x≈g(f(x)), where ≈ in this paper signifies some sort of measure of semantic similarity. The idea is that if f and g are deep learning models, then it is unlikely $g=f^{-1}$ but x≈g(f(x)), thus leading to similar semantics with different text. We utilized the “Helsinki-NLP/opus-mt-en-de” [37] and “Helsinki-NLP/opus-mt-de-en” [37] models for back-translation; we translated English text to German and back to English. This process ensures the syntactical variation of the text while preserving its semantic integrity [36], generating 138,714 new samples for training. We ran this model only on the under sampled class, meaning on data samples whose label was bullying; this allowed us to produce more bullying data samples that we could integrate into the original dataset to have a more balanced dataset. We ran this model on all the bullying samples in batches.

#### 3.4.2. Paraphrasing

Paraphrasing is the modification of a text that keeps the same meaning of the original text. Generative models like T5 can be used for paraphrasing generation. We employed a T5 [23] model trained on paraphrasing tasks to reformulate texts. T5 is an encoder–decoder model that is very similar to the original transformer proposed in “attention is all you need” [38], with a few architectural modifications. Furthermore, T5 is pre-trained on various NLP downstream tasks. We used it for paraphrasing by either condensing them or altering their wording, thereby producing 92,476 new samples. We ran this model on the original dataset three times in batches, consequently not only paraphrasing the original positive samples three times, but also paraphrasing paraphrased examples twice or once depending on the iteration in which it was produced. These augmentation techniques effectively diversified our dataset, facilitating a more balanced training environment. The balance of the dataset ended up being 43/57, still in favor of negative samples.

#### 3.4.3. Model Architecture and Training

In designing the architecture for bullying detection, we prioritized transformer-based encoders for their demonstrated efficacy in generating rich textual representations, simplifying tasks like text classification. Our architecture comprised a transformer-based encoder as the foundational element. We took the text representing a comment in the dataset, and we fed it to a tokenizer to tokenize the text. The tokenizer assigns a positive natural number to each token depending on its index on the vocabulary. The vocabulary can be seen as a function V, whose domain is pre-defined to be the set of tokens of the language you are considering. After tokenization, we get a vector of positive natural numbers that we then feed to the transformer-based encoder. This encoder maps each natural number to a real valued vector of size 512 for small encoders and 768 for large encoders. Consequently, a linear classification head is used to compute the logits, which are normalized using the Softmax function. We use cross-entropy loss as our loss function, meaning we did something similar to logistic regression on the encoded representations, using the softmax function as our probability function instead of the sigmoid function. The mathematical formulation of the aforementioned process is discussed below. Let
(1)v={x|xisatokeninlanguageL}
(2)V:v→{0,1,…,|v|−1}
(3)$$Tok:(x_{1},\ldots ,x_{n})\;\mapsto \;(V(x_{0}),V(x_{1}),\ldots ,V(x_{n}),V(x_{n+1})))$$

The tokenizer transforms a string of tokens (here represented as a tuple of tokens) to a vector of natural numbers; it also adds an integer representing the starting token and one representing the ending token. Similarly, let
(4)$$e_{i}:\left(v\left(x_{1}\right),\ldots ,v\left(x_{n}\right)\right)\to R^{512}$$
(5)$$(v\left(x_{1}\right),\ldots ,v\left(x_{n}\right))=\psi$$
(6)$$E:(v\left(x_{1}\right),\ldots ,v\left(x_{n}\right))\;\mapsto \;(e_{1}(\psi ),\ldots ,e_{n}(\psi ))$$
where $e_{1}\left(\psi \right)$ represents the string, meaning the encoding of the starting token is used to represent the whole sentence. This encoding has information from the whole string since it is a transformer-based encoder that computes each vector based on attending every other member of the sentence. This vector is then used for classification, where $C:R^{512}\to R^{2}$, and
(7)C(x)=Wx+b
where W is 2 × 512, b is a vector of size 2, and C gives the logits. We pass the output through a softmax function. In this case, the softmax is $Sft:R^{2}\to R^{2}$, where
(8)$$Sft\left(x\right)=\left(\frac{e^{x_{1}}}{\left(e^{x_{1}}+e^{x_{2}}\right)},\frac{e^{x_{2}}}{e^{x_{1}}+e^{x_{2}}}\right).$$

The forward pass is given as:(9)$$Sft\circ C\circ I\circ E\circ Tok\left(x_{1},\ldots ,x_{n}\right)=\left(P\left(\neg bullying||x\right),P\left(bullying|x\right)\right)$$
where $x=x_{1},\ldots ,x_{n}$ and I returns the first column of a matrix. Let
(10)Sft∘C∘I∘E∘Tok=M

Finally, we pass this result to the binary cross-entropy loss function:(11)$$L\left(y,x\right)=-\frac{1}{N}\sum_{i=1}^{N}\left(y_{i}log\left(M\left(x_{i}\right)\left[1\right]\right)+\left(1-y_{i}\right)log\left(M\left(x_{i}\right)\left[0\right]\right)\right)$$
where $y_{i}$ is the class index of sample i, and $x_{i}$ is the text of sample i.

We explored various encoders, including BERT [20], T5-base [23], MobileBERT [24], T5-small [23]. and SimCSE [25], to identify the optimal balance between performance and resource efficiency, particularly for deployment within a web-based system. The models were pre-trained on the augmented Kaggle dataset [22] using cross-entropy loss, a learning rate of 5e-5, and an Adam optimizer [39]. An example of the model pipeline is represented by Figure 3.

#### 3.4.4. Fine Tuning and Comparative Analysis

Following pre-training, models were fine tuned using the KIDS-cyberbullying dataset derived from children’s responses to the cyberbullying scenarios presented by our virtual companion. This dataset was similarly imbalanced, predominantly featuring instances of bullying recognition. To enhance the dataset, we again resorted to data augmentation, incorporating selected negative samples from the Kaggle dataset and employing paraphrasing using T-5 [23] to expand our data. After dividing the augmented dataset into training and testing sets, we fine tuned our most effective models for final evaluation. Our findings were juxtaposed with those of state-of-the-art models, notably a configuration utilizing RoBERTa [21] as the encoder and an long short-term memory (LSTM) as the classification layer, to gauge the relative performance and effectiveness of our approach. One can see a flowchart of our pipeline in Figure 4.

## 4. Results and Discussion

### 4.1. Evaluation Metrics and Testing Protocol

The performance of our models was rigorously evaluated on two distinct datasets: a test set of the Kaggle cyberbullying dataset (KDP) and an augmented KIDS-cyberbullying dataset derived from interactions with children on the autism spectrum. The models were assessed using a comprehensive suite of metrics, including accuracy and F1-score, to ensure a holistic evaluation of their performance in both datasets.

### 4.2. Model Performance on Kaggle Dataset

A diverse array of model configurations was tested on the KDP, encompassing variations in backbone architectures (T5-base, T5-small, MobileBERT, and BERT-base), data augmentation practices, classification heads (linear layer vs. LSTM), and encoder weight optimization strategies (frozen vs. trainable weights). Additionally, the efficacy of contrastive learning was explored in this context.

The standout models, namely T5-base, MobileBERT, and T5-small, were distinguished by their utilization of full data augmentation, a linear classification head consisting of multi-layer perceptron (MLP), and a trainable backbone, underscoring the pivotal role of data augmentation in enhancing model generalizability.

As seen in Table 1, the highest accuracy that was achieved on the KDP dataset was 98.9% with the RoBERTa-based model [40], 1.8% higher than the 96.99% accuracy of the T5-base model. Close behind, the T5-small model demonstrated a promising accuracy of 96.76%, highlighting its potential suitability for web-based applications due to its balance between performance and computational efficiency. Although the RoBERTa-based [40] model achieved a slightly higher accuracy than the T5-base and T5-small models, the sizes of T5-base, T5-small, and MobileBERT models are much smaller than the number of parameters in the RoBERTa network, as shown in Table 2. Therefore, we deployed the fast and lightweight T5-small model in our web-based online cyberbullying detection system. MobileBERT and T5-small were able to be run on mobile devices in our web-based platform.

### 4.3. Fine Tuning and Performance on the KIDS-Cyberbullying Dataset

The T5-small and MobileBERT models, identified as optimal balance of performance and efficiency through initial testing, were subsequently fine tuned on the KIDS-cyberbullying dataset, which consisted of 255 samples. T5-small encoder is a variant of the transformer encoder that uses only 6 encoders and where the embedding dimension is of size 512. Furthermore the number of attention heads is 8. The architecture can be seen in Figure 5. MobileBERT is a variant of the BERT architecture where the embedding dimension is of size 512 instead of 768, and where there is a bottleneck linear layer that reduces the dimension to 128. Furthermore, it only uses 4 attention heads. The number of encoders is 24. A figure of the architecture can be seen in Figure 6. A train–test split ratio of 0.8/0.2 was employed for this phase. For comparative analysis, the RoBERTa plus LSTM model, Bert, and T5-base were also evaluated on the KIDS-cyberbullying dataset. The fine-tuning process yielded an impressive accuracy of 94.1% on the test data for the T5-small, T5-base, and MobileBERT models. In stark contrast, the RoBERTa-based model [40] significantly underperformed, achieving an accuracy of merely 41.2% on the test data, indicative of substantial misclassification issues. These results are shown in Table 3. A confusion matrix of the results of the T5-small model is shown in Figure 7. Moreover, the accuracy on the training data and the loss on the test data on the KIDS-cyberbullying dataset can be seen in Figure 8. We also performed a time analysis where we measured the inference time of each model. This can be seen in Figure 9. We took 20 measurements for the time it took each model to infer on the test data. We took the mean of the 20 measurements of each model and then plotted the accuracy vs. mean time in seconds. Each data point represents a model, its y component represents the accuracy of the model on the test set, and its x component the mean time out of 20 measurements on the test set of the model. T5-small achieved the best mean inference time, followed by MobileBERT, which is expected, as these two are the smallest models. T5-small and MobileBERT both attained the best accuracy, which indicates that these models meet our web-based real-time cyberbullying detection requirement.

This study achieved impactful results in the development of lightweight models such as MobileBERT and T5-small. Based on these experiments, the lightweight models of T5-small and MobileBERT are integrated into our web-based system for fast real-time cyberbullying detection. This integration marks a significant step towards automating the detection of bullying recognition in the context of cyberbullying detection education. The promising outcomes of this research lay the groundwork for further advancements in the field of cyberbullying detection.

### 4.4. The Privacy, Safety, and Well-Being of the Participants

To ensure the privacy, safety, and well-being of the children, our AI companion interacts with participants only by asking for their responses after the cyberbullying scenario. This means that the AI companion is designed to engage with the children in a controlled and limited manner. It asks specific questions related to the cyberbullying scenario they have just experienced, ensuring that the interaction is focused and relevant.

Beyond this interaction, the design of our framework prevents the AI companion from any unnecessary interaction with the participants. This precaution is in place to avoid any potential breaches of the children’s privacy, safety, and well-being. By limiting the scope of the AI companion’s interaction, we minimize the risk of exposing children to any unintended or harmful content.

After receiving their responses, our system presents educational videos and materials based on the outcome of the proposed automatic cyberbullying detection method. This means that once the children have provided their feedback, the system uses this information to determine which educational resources would be most beneficial for them. These resources are tailored to address the specific issues and lessons highlighted by the cyberbullying detection method.

As illustrated in the flowchart of the sequence of events in Figure 5 of the paper, this process ensures that the educational content is relevant and supportive, helping children to understand and cope with cyberbullying effectively. The flowchart provides a visual representation of this sequence, showing each step from the initial interaction with the AI companion to the presentation of the educational materials. This structured approach helps to reinforce the learning experience and ensures that the children receive the guidance they need in a safe and supportive environment.

### 4.5. Improvement and Future Work

We have deployed the proposed cyberbullying detection and education framework in real time. In the near future, we plan to conduct additional educational sessions using our method. We will collect data from these upcoming sessions to expand our dataset and further improve the accuracy of the proposed technique. Looking ahead, we are committed to expanding the scope of our research to include more robust models and to undertake a multi-modal analysis approach. This expanded approach will not be limited to textual data; instead, it will incorporate audio and video data to enhance the predictive capabilities of our models. The inclusion of these additional data types is anticipated to improve the accuracy and reliability of bullying detection. Moreover, our future research endeavors will also focus on the detection of incidents where private protected information is disclosed within cyberbullying contexts. This aspect is particularly crucial, as the release of personal information can have devastating consequences for the victims. The ability to accurately identify such incidents will be a significant addition to our cyberbullying detection framework. The importance of understanding and addressing cyberbullying cannot be overstated, particularly in today’s digital age where such behaviors are increasingly prevalent. This is especially true for vulnerable populations, including children on the autism spectrum, who may be at a higher risk of experiencing cyberbullying [46,47]. Through our ongoing research and development efforts, we aim to contribute significantly to the protection of these individuals by providing robust, effective tools for cyberbullying detection and prevention.

## 5. Conclusions

Social media platforms and online gaming sites serve as significant mediums for peer interaction and social growth among adolescents. However, they also present inherent risks to health and safety. Addressing cyberbullying within these platforms is essential for fostering the healthy social development of adolescents. Cyberbullying correlates with negative mental health outcomes in adolescents, such as anxiety, depression, academic decline, and an increased suicide risk. While cyberbullying affects all adolescents, those with disabilities face heightened susceptibility and are more likely targets. Our research addresses these challenges by introducing a personalized online virtual companion guided by Artificial Intelligence (AI). This web-based virtual companion interacts with adolescents to help them detect cyberbullying instances. Specifically, in a virtual scenario, an adolescent with ASD encounters a cyberbullying situation and is queried by the AI virtual companion about their perception of cyberbullying. To enable real-time feedback for the virtual companion, we have implemented lightweight cyberbullying models using the T5-small and MobileBERT networks. Our experimental findings demonstrate comparable results to state-of-the-art methods, despite employing a compact architecture.

## Figures and Tables

**Figure 1 sensors-24-03875-f001:**
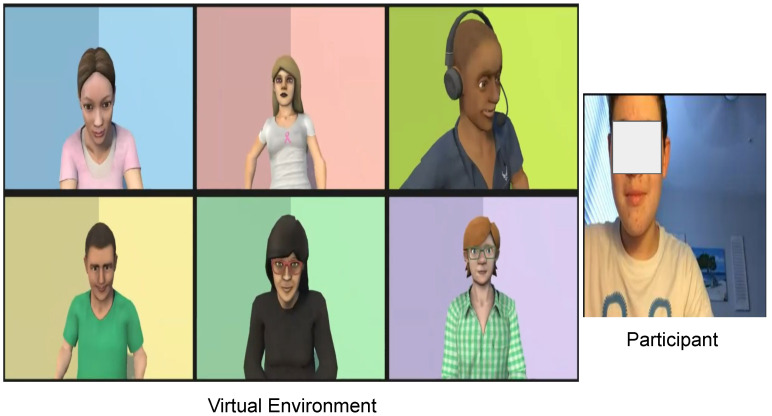
Participant interacting with the virtual environment.

**Figure 2 sensors-24-03875-f002:**
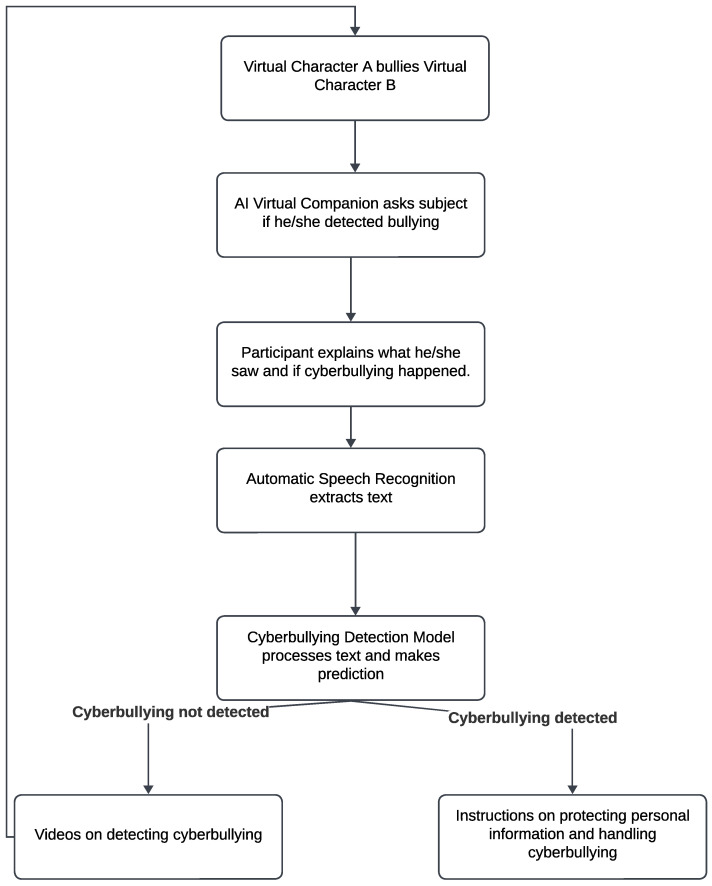
Interaction of the subject with the virtual environment pipeline.

**Figure 3 sensors-24-03875-f003:**
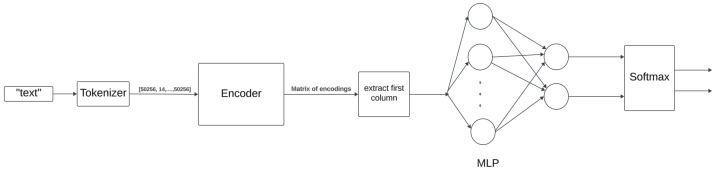
Flowchart of model pipeline.

**Figure 4 sensors-24-03875-f004:**
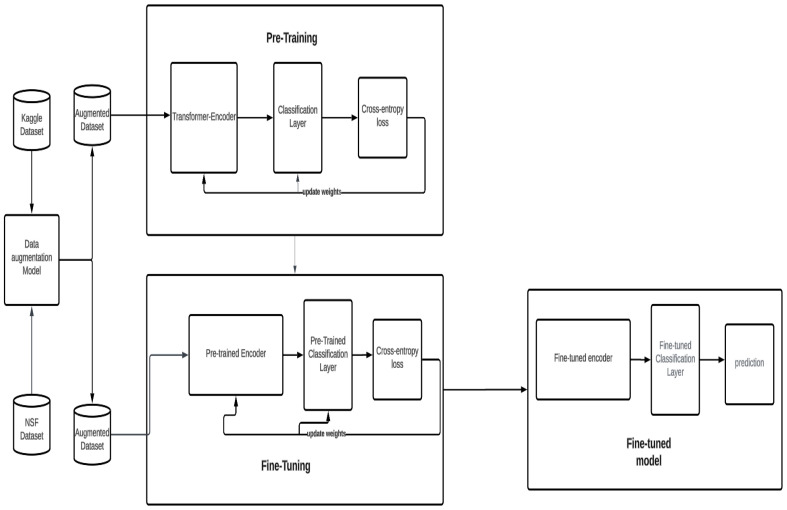
Flowchart of our cyberbullying detection.

**Figure 5 sensors-24-03875-f005:**
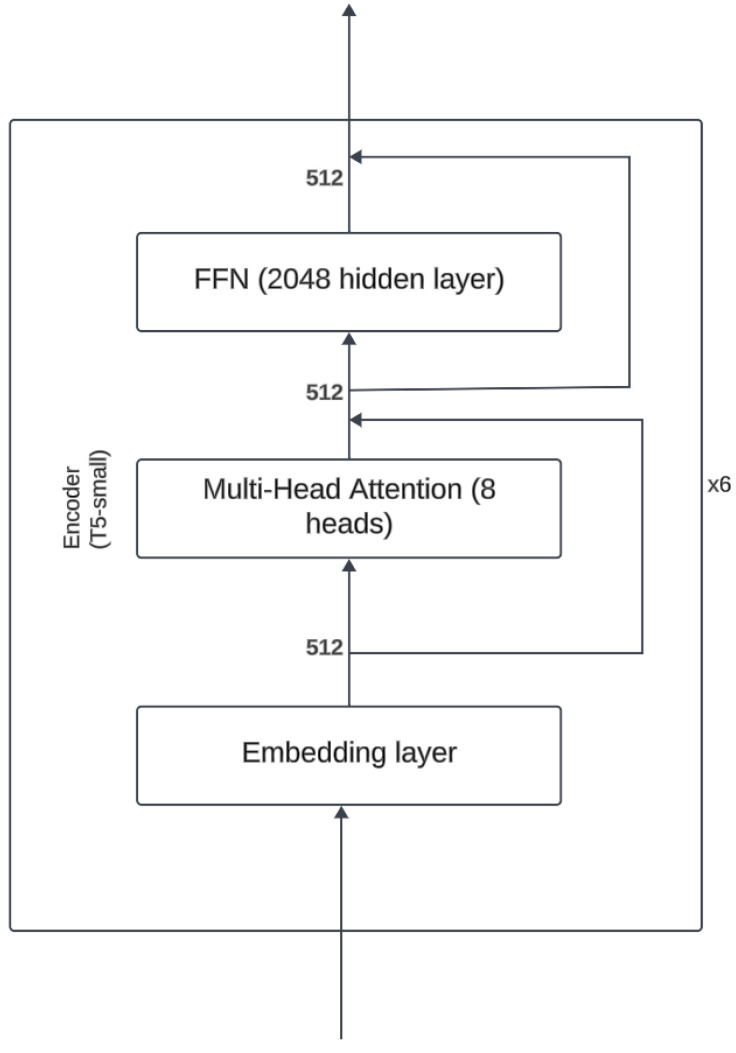
T5-small encoder architecture.

**Figure 6 sensors-24-03875-f006:**
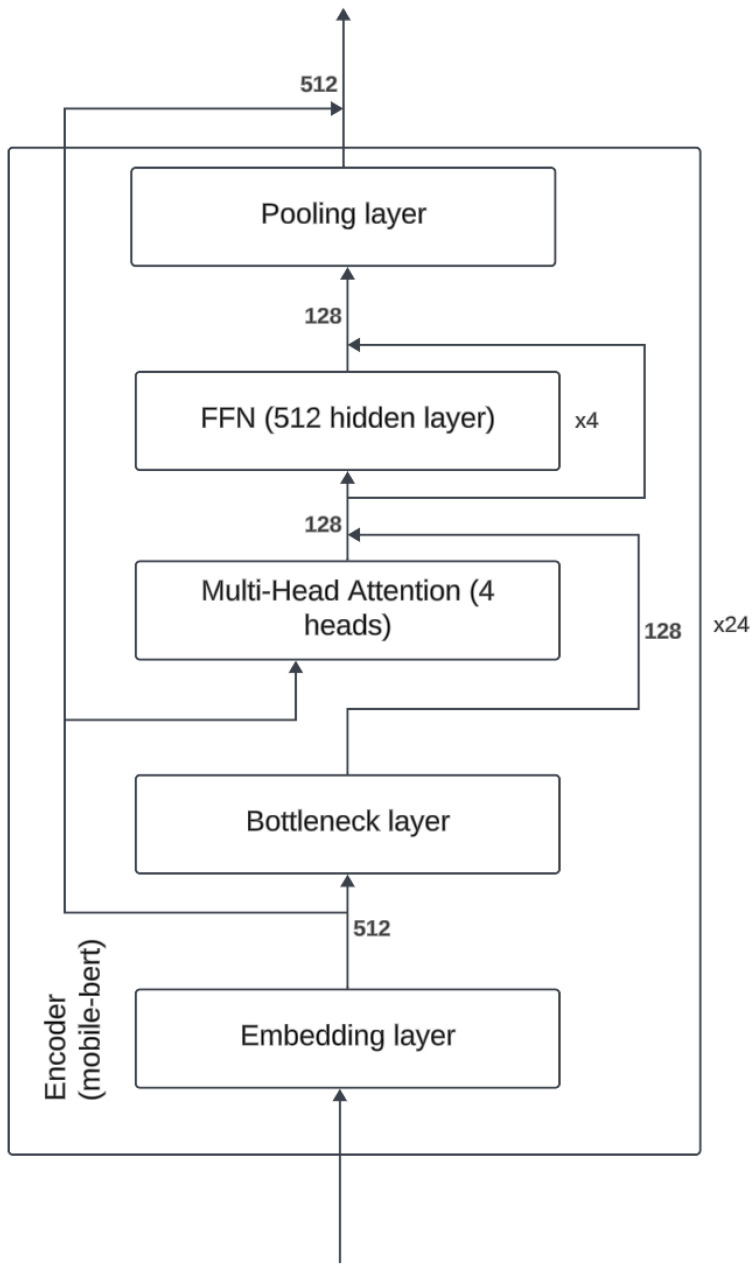
MobileBERT architecture.

**Figure 7 sensors-24-03875-f007:**
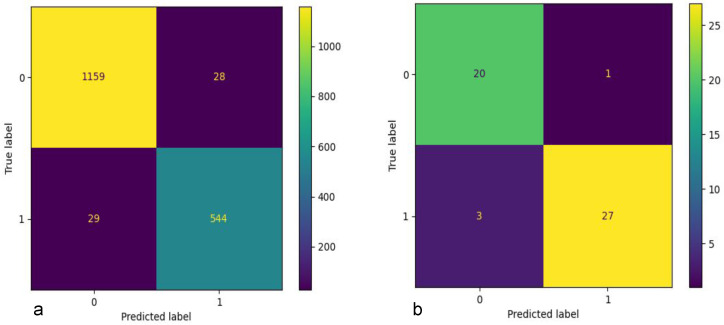
(**a**) The confusion matrix of T5-small using the Kaggle dataset, and (**b**) the confusion matrix of T5-small KIDS-cyberbullying dataset.

**Figure 8 sensors-24-03875-f008:**
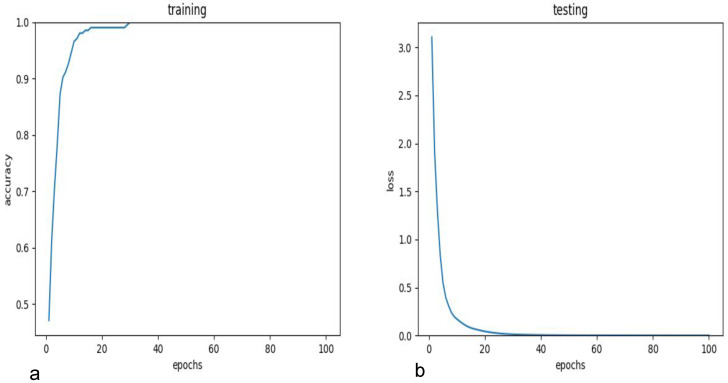
(**a**) The accuracy of T5-small using the KIDS-cyberbullying dataset, and (**b**) the loss of T5-small using the KIDS-cyberbullying.

**Figure 9 sensors-24-03875-f009:**
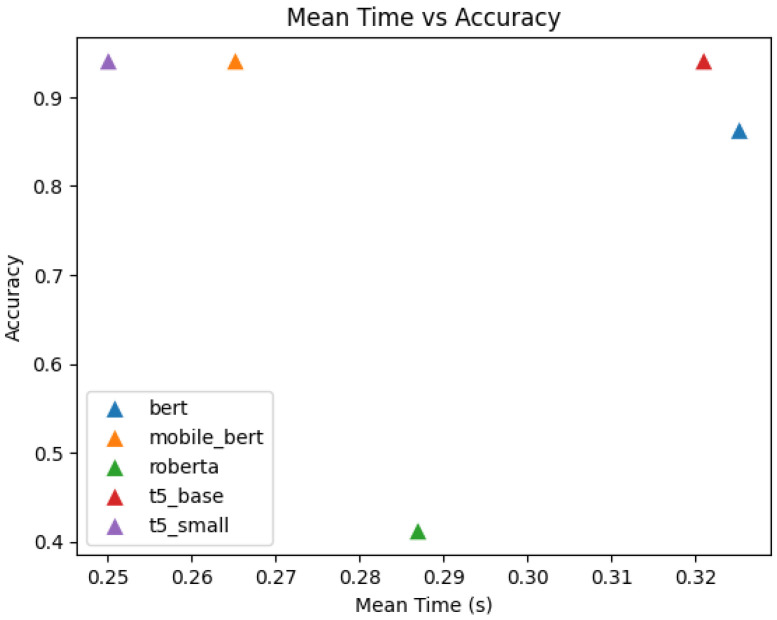
Time when compared to the accuracy of each model for inference.

**Table 1 sensors-24-03875-t001:** The performance metrics for the evaluated models and comparison to the state-of-the-art methods.

Model	Parameters/Details	Accuracy/F1-Score
SVM [40]	Not pre-trained, word2vec features	Acc: 0.394, F1-score: 0.479
Linear regression [40]	Bag of words features	Acc: 0.809, F1-score: 0.766
BiSTM [41]	Word2Vec features, Facebook dataset	Acc: 0.821
LSTM [42]	TF-IDF features, Twitter 47,694 dataset	F1-score: 0.920
CNN [33]	Glove features, Twitter- 69,876 dataset	Acc: 0.930
BERT [43]	Word2Vec features, Facebook-5000 dataset	F1-score: 0.928
RNN [44]	FastText features, Aahaber dataset	Acc: 0.935
GRU [45]	FastText features, TTC-3600 dataset	F1-score: 0.960
RoBERTa [40]	Full augmentation, LSTM, Not frozen, KDP dataset	Acc: 0.988, F1-score: 0.988
BERT	No augmentation, MLP, Not frozen, SimCSE, KDP dataset	Acc: 0.871
BERT	No augmentation, MLP, Not frozen, KDP dataset	Acc: 0.873
BERT	No augmentation, MLP, frozen, SimCSE, KDP dataset	Acc: 0.928, F1-score: 0.683
MobileBERT	No augmentation, MLP, Not frozen, KDP dataset	Acc: 0.955, F1-score: 0.932
MobileBERT	Full augmentation, MLP, Not frozen, KDP dataset	Acc: 0.959, F1-score: 0.842
T5-base	Full augmentation, LSTM, Not frozen, KDP dataset	Acc: 0.948, F1-score: 0.920
T5-base	paraphrasing, MLP, Not frozen, KDP dataset	Acc: 0.964, F1-score: 0.855
T5-base	No augmentation, MLP, Not frozen, KDP dataset	Acc: 0.965, F1-score: 0.947
T5-base	Full augmentation, MLP, Not frozen, KDP dataset	Acc: 0.969, F1-score: 0.954
T5-small	Full augmentation, MLP, Not frozen, KDP dataset	Acc: 0.968, F1-score: 0.950

**Table 2 sensors-24-03875-t002:** Comparison of the number of parameters of the models.

Model	Number of Parameters
RoBERTa	124,651,808
T5-base	109,630,082
BERT	109,483,778
T5-small	35,331,842
MobileBERT	24,582,914

**Table 3 sensors-24-03875-t003:** The performance metrics for the evaluated models on the KIDS-cyberbullying.

Model	Accuracy	Recall	Precision	F1-Score
RoBERTa	0.412	0.0	0.0	0.0
BERT	0.863	0.867	0.897	0.881
Contrastive Bert	0.882	0.866	0.929	0.896
T5-base	0.941	0.967	0.935	0.951
MobileBERT	0.941	0.967	0.935	0.951
T5-small	0.941	0.9	1	0.947

## Data Availability

The original contributions presented in the study are included in the article, further inquiries can be directed to the corresponding author. The details of the publicly available dataset used in this work are provided in the paper. However, the KIDS-cyberbullying data of participants will not be made public, as the IRB requires explicit permission for access to each individual’s data.

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
