# Peer review of "Using AI-Based Virtual Companions to Assist Adolescents with Autism in Recognizing and Addressing Cyberbullying"

_sensors, 2024, doi:10.3390/s24123875_

Round 1
Reviewer 1 Report
Comments and Suggestions for Authors
The authors describe in the article a proposed approach to identifying cyberbullying in a textual context. The major advantage of the proposed one is the high speed of operation and the ability to be used in web systems.
The idea of ​​the article is relevant and interesting.
I can highlight the following article problems:
1. It is not clear to me how the authors interact with the subject in the proposed virtual environment. What functions does the environment provide? What should the test taker do to get the result or textual data for analysis? The authors described these aspects superficially.
2. The section describing the methodology is the weakest point of the article. I would like to see more examples and implementation details of the proposed approach. What does a pipeline look like for processing data from input to output? What parameters and features does each pipeline block have? I advise authors to add examples of input data, detailed descriptions of data transformations, and an example of the result for each pipeline block.
3. The section describing the experimental results lacks an analysis of the time spent on data analysis between the proposed approach and the existing ones. Since the authors claim higher operating speed, it is logical for the authors to prove the speed advantage of the proposed approach. It is possible that I did not look carefully, but I did not see a comparison of the quality of the proposed approach with analogues. How many does “acceleration” of analysis affect the quality of the result?
4. The most important problem of the article is the lack of novelty. Using existing libraries in the right order and with the right parameters is not a scientific novelty. The authors should highlight the novelty of the proposed approach. How does the proposed approach differ from existing ones except for the use of “lighter” models? What new did the authors bring to their field of science?
I recommend that the authors seriously rework the article considering the comments.
Author Response
Responses to the reviewers:
Reviewer_1:
Comment _1 :It is not clear to me how the authors interact with the subject in the proposed virtual environment. What functions does the environment provide? What should the test taker do to get the result or textual data for analysis? The authors described these aspects superficially.
Re: The details of the virtual environment and the data collection is discussed in detail in section 3.1 of the paper.
Comment _2 :The section describing the methodology is the weakest point of the article. I would like to see more examples and implementation details of the proposed approach. What does a pipeline look like for processing data from input to output? What parameters and features does each pipeline block have? I advise authors to add examples of input data, detailed descriptions of data transformations, and an example of the result for each pipeline block.
Re: We have updated the methodology section of the paper having more examples and implementation details of the proposed approach.
Comment _3 :The section describing the experimental results lacks an analysis of the time spent on data analysis between the proposed approach and the existing ones. Since the authors claim higher operating speed, it is logical for the authors to prove the speed advantage of the proposed approach. It is possible that I did not look carefully, but I did not see a comparison of the quality of the proposed approach with analogues. How many does “acceleration” of analysis affect the quality of the result?
Re: We have performed the inference time analysis of the models, and discussed the results in the section 4.3 of the paper. The graphical representation of the time analysis is shown in Figure 7.
Comment _4 :The most important problem of the article is the lack of novelty. Using existing libraries in the right order and with the right parameters is not a scientific novelty. The authors should highlight the novelty of the proposed approach. How does the proposed approach differ from existing ones except for the use of “lighter” models? What new did the authors bring to their field of science?
Re: In this paper, we present, to the best of our knowledge, the first web-based AI companion designed to help kids not only detect cyberbullying but also learn how to handle such scenarios. Our lightweight, real-time cyberbullying detection method teaches kids how to protect their private data. Our focus on models, such as T5-small and MobileBERT, for real-time deployment in a web-based system is practical and addresses the computational constraints of such systems. We address the important problem of identifying whether a participant detects cyberbullying in a virtual environment by formulating the issue as a text-based cyberbullying detection problem. Although the participant’s response may not technically match instances of bullying speech, we posit that their description of the observed virtual environment can serve as input for the cyberbullying detection model. This framework also minimizes random guesses by the participant in cyberbullying detection. This innovative problem formulation, tailored for a unique and crucial application, constitutes a significant contribution to the scientific community. This is the first paper to introduce a lightweight cyberbullying detection method that can be used in web-based, real-time cyberbullying detection and prevention education.

Reviewer 2 Report
Comments and Suggestions for Authors This article addresses a crucial issue by introducing a personalized online virtual companion guided by Artificial Intelligence to assist adolescents, particularly those on the autism spectrum, in detecting cyberbullying. The authors' focus on developing lightweight models, such as T5-small and MobileBERT, for real-time deployment in a web-based system is practical and addresses the computational constraints of such systems. While the authors have acknowledged the limited size and imbalance of the KIDS-cyberbullying dataset, they should discuss strategies for expanding and diversifying this dataset in future work. A larger and more balanced dataset could further improve the performance and generalizability of their models. Given the involvement of adolescents, particularly those with ASD, in their study, the authors should discuss the ethical considerations and safeguards they have implemented to ensure their privacy, safety, and well-being. The authors provides a brief description of the virtual companion's role, but it lacks specifics on its functionalities and interaction design. A more detailed explanation of how the virtual companion interacts with adolescents, the types of scenarios presented, and the feedback mechanisms employed would be beneficial. The article primarily focuses on textual information and lacks exploration of other modalities such as visual and auditory cues. Expanding the evaluation to incorporate these modalities would provide a more comprehensive understanding of the virtual companion's effectiveness. Although the approach seems promising and innovative, it has several shortcomings. Major revisions are needed before publication.
Author Response
Responses to the reviewers:
Reviewer_2:
Comment _1 :This article addresses a crucial issue by introducing a personalized online virtual companion guided by Artificial Intelligence to assist adolescents, particularly those on the autism spectrum, in detecting cyberbullying. The authors' focus on developing lightweight models, such as T5-small and MobileBERT, for real-time deployment in a web-based system is practical and addresses the computational constraints of such systems. While the authors have acknowledged the limited size and imbalance of the KIDS-cyberbullying dataset, they should discuss strategies for expanding and diversifying this dataset in future work. A larger and more balanced dataset could further improve the performance and generalizability of their models. Given the involvement of adolescents, particularly those with ASD, in their study, the authors should discuss the ethical considerations and safeguards they have implemented to ensure their privacy, safety, and well-being. The authors provides a brief description of the virtual companion's role, but it lacks specifics on its functionalities and interaction design. A more detailed explanation of how the virtual companion interacts with adolescents, the types of scenarios presented, and the feedback mechanisms employed would be beneficial. The article primarily focuses on textual information and lacks exploration of other modalities such as visual and auditory cues. Expanding the evaluation to incorporate these modalities would provide a more comprehensive understanding of the virtual companion's effectiveness. Although the approach seems promising and innovative, it has several shortcomings. Major revisions are needed before publication.
Re: We have deployed the proposed cyberbullying detection and education framework in real-time. This summer, we plan to conduct additional educational sessions using our method. We will collect data from these upcoming sessions to expand our dataset and further improve the accuracy of the proposed technique.
To ensure the privacy, safety, and well-being of the children, our AI companion interacts with participants only by asking for their responses after the cyberbullying scenario. Beyond this interaction, the design of our framework prevents the AI companion from breaching the privacy, safety, and well-being of the children. After receiving their responses, our system presents educational videos and materials based on the outcome of the proposed automatic cyberbullying detection method, as illustrated in the flowchart of the sequence of events in Figure 2 of the paper.
Regarding the virtual companion's role and its interaction with the participants, we have presented the details of the virtual environment and the sequence of events in Section 3.1 of the paper.
To expand the proposed work by incorporating additional modalities such as visual and auditory data, we plan to compile our own multimodal cyberbullying database in the near future. The extensions being proposed are also discussed in Section 4.3 of the paper. To the best of our knowledge, there is no published multimodal cyberbullying dataset that meets our specific requirements. The datasets available are mostly collected from social media platforms such as Facebook (Meta), Twitter (X), and Instagram. Our analysis found these datasets to be infeasible for our needs, as their image and video content relate to cyberbullying scenarios within a social media context. For example, the samples in these datasets include images of posts from the aforementioned social media platforms and text data related to those posts. In our case, however, we are more interested in capturing facial expressions and associated audio information when the participant explains the cyberbullying scenario.

Round 2
Reviewer 1 Report
Comments and Suggestions for Authors
Can be accepted
Author Response
We appreciate the reviewer's prior feedback that we feel greatly improved the paper. The reviewer accepted our changes as meeting all needed edits, so we have no explicit response associated with this review.
Reviewer 2 Report
Comments and Suggestions for Authors
In this version the authors have provided interesting elements of answers to my various questions.
However, these answers are not sufficiently integrated and developed in the paper. Authors should add further dedicated paragraphs to detail these concepts.
Otherwise the proposed approach is well structured and argumented.
Author Response
Responses to Reviewer_2:
Comment : In this version the authors have provided interesting elements of answers to my various questions.
However, these answers are not sufficiently integrated and developed in the paper. Authors should add further dedicated paragraphs to detail these concepts.
Otherwise the proposed approach is well structured and argumented.
Re: We have addressed the important concerns raised by the reviewer by forming separate sections in the revised manuscript. The improvement of the proposed technique in the near future, through the compilation of more data samples from our upcoming sessions, is discussed in a newly formed section 4.5. Additionally, section 4.5 also details the future expansion of the proposed method by integrating other modalities such as facial expressions and audio information. This will be achieved by compiling our own multimodal cyberbullying detection dataset using our proposed web-based real-time system.
To provide a detailed discussion on the privacy, safety, and well-being of the participants in our proposed web-based real-time cyberbullying detection system, we have added a separate section, section 4.4, in the revised manuscript. This section covers the limited interaction of the AI companion with the children after a virtual cyberbullying scenario and emphasizes the protective measures integrated into our proposed method to ensure a safe and supportive learning environment.